# Prevalence and Determinants of Excessive Screen Viewing Time in Children Aged 3–15 Years and Its Effects on Physical Activity, Sleep, Eye Symptoms and Headache

**DOI:** 10.3390/ijerph20043449

**Published:** 2023-02-16

**Authors:** Shilpa Jain, Shreya Shrivastava, Aditya Mathur, Deepali Pathak, Ashish Pathak

**Affiliations:** 1Department of Pediatrics, RD Gardi Medical College, Ujjain 456010, India; 2Health Systems and Policy, Department of Global Public Health, Karolinska Institutet, 17176 Stockholm, Sweden

**Keywords:** screen view time, children, risk factors, India

## Abstract

Screen viewing time is the total time spent by a child on any digital/electronic device. The objective of the present study was to determine the prevalence and predictors of excessive screen viewing time in children in Ujjain, India. This cross-sectional, community-based study was conducted through a house-to-house survey using the three-stage cluster sampling method in 36 urban wards and 36 villages of Ujjain District, India. Excessive screen viewing time was defined as screen viewing for >2 h/day. The prevalence of excessive screen viewing time was 18%. Risk factors identified using the multivariate logistic regression model were age (OR: 1.63, *p* < 0.001); mobile phone use before bedtime (OR: 3.35, *p* = 0.004); parents’ perception about the child’s habituation to screen time (OR: 8.46, *p* < 0.001); television in the bedroom (OR: 35.91, *p* < 0.001); morning mobile screen viewing time (OR: 6.40, *p* < 0.001); not reading books other than textbooks (OR: 6.45, *p* < 0.001); and lack of outdoor play for >2 h (OR: 5.17, *p* < 0.001). The presence of eye pain was a protective factor for excessive screen viewing time (OR: 0.13, *p* = 0.012). This study identified multiple modifiable risk factors for excessive screen viewing time.

## 1. Introduction

Screen viewing time (SVT), or digital/screen exposure, is the total time spent by an individual viewing or using any digital or electronic device, such as a television (TV), smart phones, tablets, or computers [1]. According to the American Academy of Pediatrics (AAP) guidelines, children aged less than 2 years should not have any SVT, and ≥1 h/day of SVT is considered excessive among children aged 2–5 years [2]. Recent data suggest that children begin using online platforms at an early age in high-income settings [1,2]. The effects of prolonged SVT and its physical, psychosocial, behavioral, and long-term outcomes are gradually becoming apparent [3,4,5,6]. The most studied effect of prolonged SVT is the reduced physical activity (PA) (lower than the recommended limit) of children and adolescents. The World Health Organization (WHO) has recommended at least 2 h of PA per day for children and adolescents [7]. In many high-income countries, such as the United States of America (USA) and Japan, the recommended limit set by the WHO on PA per day has not been achieved [2,4]. Long SVT is associated with low PA and long sedentary periods [4], both of which are associated with obesity [4]. In the USA, most preschool children spend at least 4 h viewing screens daily [2]. Almost all school-going children in the USA watch TV, and one-third of young children play games on computers or electronic devices [8].

Increased SVT is associated with adult-onset diseases associated with a sedentary lifestyle in high-income countries [9,10,11]. watching TV and the use of smart phones and tablets have been shown to significantly reduce PA, cause dry eyes, and reduce school performance [12]. Almost 90% of the studies included in a systematic review indicated that increased SVT is associated with adverse sleep outcomes [13]. Sitting for lengthy periods of time in a fixed posture in front of any electronic device, such as computer terminals, may result in neck pain and headache [14].

India is the largest and fastest-growing market for digital consumers, with 560 million internet subscribers and 1.2 billion mobile phone subscriptions in 2018 [15]. Despite the increase in exposure to digital screens in India, no recommendations are currently available on screen viewing and its effects. Moreover, no studies have comprehensively evaluated the known risk factors for excessive SVT. Therefore, the objective of the present study was to determine the prevalence and predictors of excessive SVT in children and its effect on physical/outdoor activity, sleep, and the prevalence of eye symptoms and headache in children in Ujjain District, Madhya Pradesh, India.

## 2. Materials and Methods

This cross-sectional study, using a house-to-house survey, was conducted from May to August 2021 in 36 urban wards and 36 villages of Ujjain District. Ethical approval of the study was obtained from the Institutional Ethics Committee of RD Gardi Medical College, Ujjain (Approval Number 95/2019). Children aged 3–15 years were identified during house-to-house visits.

### 2.1. Sample Size Calculation

The modified WHO cluster sampling method was used for sample size calculation [16]. Prior to sampling, children were stratified according to age into three groups, namely 3–6 years, >6–10 years, and >10–15 years. In a study in Karamsad, Gujrat, India, 17% children had less than one hour of SVT per day [17]. According to the 2014 National Family Health Survey data for Ujjain District [18], the prevalence of appropriate duration of screen time is 17%, the conservative design effect is 2, the nonresponse rate is 10%, the estimated number of households needed to find eligible children is 0.59, the number of children per cluster is 7, and the minimum estimated sample size is 550. The estimated minimum number of households we needed to visit per age stratum was 110, with 24 clusters per stratum and 5 households per cluster. Thus, the survey comprised 72 clusters (36 urban and 36 rural).

### 2.2. Sampling Procedure

To select the clusters, Ujjain Tehsil of Ujjain District, Madhya Pradesh, India, was considered for the sampling frame. The clusters were defined in urban areas (Ujjain) at the ward and village levels in rural areas. In total, Ujjain has 54 wards, and Ujjain Tehsil has 132 villages. A total of 36 urban wards and 36 villages were randomly selected from the list of wards and villages.

In this community-based study, we selected participants at the household level using the three-stage cluster sampling method. First, all residential clusters within a radius of 0.5 km from the selected clusters were identified and numbered sequentially. Then, a census of households was conducted to identify households with children aged 3–15 years via house-to-house visits. In the second stage, 5 eligible households were randomly selected from each urban and rural cluster of urban wards and villages. As the number of eligible households had already been identified, the interviewers received a list that indicated the households to enroll. In the third stage, all eligible children of the selected households were identified, and one of them was selected randomly. Additional inclusion criteria for both clusters were (1) providing informed consent, (2) being a resident of the selected cluster for at least 6 months, and (3) being mentally capable of responding to the questionnaire. The sampling procedure is shown in Figure 1.

A questionnaire was developed through a review of the literature by two subject experts. The questionnaire was designed in both the English and Hindi languages. The parent/child was free to choose the language of the questionnaire. The questionnaire was face-validated in a pilot study conducted among 30 children. The full age range of 3–15 years was tested in the pilot study, and the questionnaire was found to be appropriate for use. The content validity index of the questionnaire was 0.87 (Appendix A, questionnaire and calculation of the content validity index). The results of the pilot study are not included in the final results. Minor changes to the questionnaire were made after the pilot study. The questionnaire comprised questions on the sociodemographic features of parents. Questions about screen viewing time and related physical, social, emotional, and behavioral domains were also included. Parents were asked to self-report the average amount of time their child spent watching TV, playing on a smart phone or tablet, using computers, playing video games, playing outdoors, and reading on a typical weekday (Monday–Friday) and a typical weekend (Saturday and Sunday), separately. Parents were handed the questionnaire during the first household visit (n = 1826). Informed consent was obtained from one of the parents, and assent to participate was obtained from children aged >7 years. In the case where the parents were not sure of their child’s screen time, they were provided additional time to observe their children’s viewing habits for one week before filling out the questionnaire. Children between 10 and 15 years of age completed their questionnaire themselves. In the subsequent household visit, the filled-in questionnaires were collected by research assistants of a nonmedical background with 10 years of research experience (Appendix A). Research assistants interviewed the parent and/or the child to fill in incomplete questionnaires.

Excessive SVT was defined as >2 h of screen activity in a day on any device, including a TV screen, computers (desktop computer and laptops), tablets such as iPads or Samsung Galaxy tablets, and mobile phones as per the Indian Academy of Pediatrics guidelines [19]. Excessive SVT was defined post hoc as the guidelines were not available at the start of the study. However, the primary outcome from the beginning of the study was defining excessive SVT.

### 2.3. Statistical Method

Descriptive statistics were used to calculate the prevalence of screen time for TV, smart devices, and computers. Differences in the screen time exposure between weekdays and weekends were assessed using paired two-tailed t tests. The outcome variable was greater than 2 h of SVT per day. Bivariate logistic regression analysis was used to assess the association of risk factors with the outcome variable. The effect of SVT on sleep, eating habits, and outdoor play was determined using bivariate logistic regression analysis. Stepwise multivariate logistic regression with backward elimination was used to develop the final model. In bivariate analyses, a *p* value of < 0.1 was considered significant for entry into the model, with excessive SVT as the outcome variable. The independent variables were girls versus boys; place of residence (urban versus rural); age in years (less than and equal to 6 versus more than 6 years); type of family (joint versus nuclear); number of family members (3 to 5 versus >5 to 9 versus more than 10); overcrowding (yes versus no); mother’s level of education (uneducated/till primary school versus up to high school versus graduate/postgraduate); mother’s occupation (unemployed versus laborer versus salaried job); father’s level of education (uneducated/till primary school versus up to high school versus graduate/postgraduate); father’s occupation (unemployed versus self-employed versus laborer versus salaried job). The independent variables for child-related health factors were night sleep time (more than or equal to 6 h versus less than 6 h); mobile phone use at bedtime (no versus yes); TV use at bedtime (no versus yes); morning mobile screen time (no versus yes); child habituated to screen (no versus yes); child read books other than school books (yes versus no); story telling by parents (yes versus no); outdoor play (yes versus no); outdoor play for more than or equal to 2 h (yes versus no); headache (no versus yes); eye pain (no versus yes); eye itching (no versus yes).

Backward elimination was repeated till the *p* value of all the predictor variables except for age and sex was less than 0.1; we designated this as the final model. Adjusted odds ratios and their respective 95% confidence intervals (CIs) were then calculated from the final model. A *p* value of <0.05 was considered significant in the final model.

## 3. Results

A total of 600 children, including 314 (52%) girls and 286 (48%) boys, were included, with a mean (± standard deviation) age of 8.82 (±3.3) years (range of 3–15 years). An approximately equal number of urban and rural children were included. The prevalence of excessive SVT (>2 h) was 18% (95% CI: 17.79–18.52%). Sociodemographic risk factors for excessive SVT are presented in Table 1.

The bivariate analysis results for the association of excessive SVT with the type of screen, its duration, and child-related health factors are presented in Table 2. Table 3 shows the results of the multivariate logistic regression model.

The factors identified to have a significant association with SVT according to the multivariate logistic regression model were age (OR: 1.63, CI: 1.36–1.91; *p* < 0.001); mobile phone use at bedtime (OR: 3.35, CI: 1.46–7.69; *p* = 0.004); parent’s perception about child’s habituation to screen viewing (OR: 8.46, CI: 2.77–25.96, *p* < 0.001); availability of a TV in bedroom (OR: 35.91, CI: 13.30–96.94; *p* < 0.001); morning mobile screen viewing time (OR: 6.40, CI: 2.76–14.84; *p* < 0.001); child not reading books other than textbooks (OR: 6.45, CI: 2.35–17.73; *p* < 0.001); less than 2 h of outdoor play in a day (OR: 5.17, CI: 2.24–11.95; *p* < 0.001). The presence of eye pain was identified as a protective factor for excessive SVT (OR: 0.13, CI: 0.02–0.68; *p* = 0.012).

## 4. Discussion

The definition of the outcome variable, excessive SVT, varies considerably in the literature. One study [20] defined it as more than 4 h in a day, while another one defined it as more than 3 h [21], and the Indian Academy of Pediatrics defined it as 2 h [19]. The American Association of Pediatrics guidelines state that children aged <2 should not have any ST, while those aged 2–5 years should be limited to 1 h [2]. Thus, the definition of excessive SVT should be kept in mind while making comparisons between studies and their generalizability and reproducibility. We therefore need a standardized definition of excessive SVT.

It was not the objective of the present study to look at the effects of the COVID-19 pandemic on SVT as our study was planned before the pandemic. However, the pandemic-induced lockdown did have an effect on the SVT of both parents and children [22,23]. An increase in TV and multimedia mobile use was observed [24]. Schools started distance learning classes, thus increasing SVT for education purposes [24]. Increased SVT leading to decreased physical activity was also observed among adolescents [25].

In the present study, the bivariate analysis showed that boys had 1.7-times-higher odds compared to girls for excessive SVT. However, the differences were not found to be statistically significant in the multivariate analysis. The results of the present study differ from those of a study conducted in New Delhi, India, which found that boys have more SVT than girls, with boys having 1.36-times-greater odds than girls [26]. According to a study from the Democratic Republic of China, the prevalence of excessive SVT is 14.7% in boys and 8.9% in girls [27]. A Malaysian study reported the mean SVT for girls and boys to be 2.8 h/day and 3.3 h/day, respectively [28].

Increasing age was found to be significantly correlated with excessive SVT (OR: 1.63, 95% CI: 1.36–1.91; *p* < 0.001) in the present study (Table 3). The results are similar to those of a study conducted in rural Western India, which showed that the odds of excessive SVT are 1.3 and 1.9 times greater in children aged 3–5 years and 5–6 years, respectively, than in children aged 2–3 years [17]. A study in China also reported that SVT among junior high school children was higher than that among elementary school children, and it decreased after 15 years of age (senior high school). The decreased SVT in Chinese senior high school children might be because of the pressure of studying and preparing for college entrance examinations, which may compel students to pay more attention and devote more time to study-related behavior, resulting in a “crowding-out effect” in the motivation crowding theory [27]. Crowding out means that when an individual is offered extrinsic incentives for certain kinds of behavior—such as promising rewards for accomplishing a task—can sometimes undermine the intrinsic motivation for performing that behavior [29].

A study in United Kingdom also showed that the time spent watching TV increased with age, and a period of an accelerated increase was observed between 12 and 30 months, with an average estimated daily TV time of 55 min at 6 months, increasing to 124 min at 36 months [1]. In the present study, we could not document the increase in SVT over time because of the observational design of the study, warranting further longitudinal studies.

In our study, excessive SVT in children was not found to have any statistically significant correlation with the type of family, overcrowding, or mother’s education and occupation (Table 1). One study used a social ecological model to explain the factors associated with digital media exposure among children [30]. Another study in Southern India reported that excessive SVT in children was not correlated with their socioeconomic status, place of residence, or mother’s education level [31]. However, a study conducted in Finland among children aged 15–16 years showed that the parents’ socioeconomic status is a risk factor for media screen exposure only among adolescent girls [32].

In the present study, the use of mobile devices before bedtime exhibited a statistically significant association with excessive SVT. A study in Tokyo, Japan, reported that children who did not have screen time before bedtime were more likely to have a normal body weight (OR: 0.73, 95% CI: 0.60–0.90), no dry eyes (OR: 1.31, 95% CI: 1.15–1.50), a better understanding of the material presented in their classes, and better academic performance (1 h to < 3 h, OR: 1.67, 95% CI: 1.41–1.98; < 1 h, OR: 2.40, 95% CI: 1.95–2.96) than those who had screen time before bedtime [33]. 

A study from China among children aged from 3 to 6 years indicated a J-shaped (positive nonlinear) association between TV viewing time and the risk of sleep disorder, with a threshold of 1 h/day. For each 1 h/day increment in TV viewing time over the threshold, the risk of sleep disorder increased by 12.35% (95% CI: 1.87–23.92%) [6]. The association between reduced night-time sleep and excessive SVT may be partially explained by the displacement hypothesis [34]. Spending more time on viewing screens reduces the time spent on other activities, such as sleep. Furthermore, the use of screen-based devices (particularly at night) exposes children to blue light, which delays sleep onset and reduces sleep quality [34].

For children with TV sets in their bedroom, the odds of >2 h of SVT was statistically significantly associated with excessive SVT. The American Academy of Sleep Medicine guidelines—endorsed by the AAP—recommend that children’s bedrooms should be free of any screen-based device and that children should not have access to any screen-based device 30 min before bedtime [35]. Bedroom media also pose the risk of obesity and video game addiction [35]. Children with bedroom media are also likely to be exposed to media violence [36]. A study in Sweden reported that shorter sleep duration was associated with having a TV set in the bedroom, spending more than 2 h/day watching TV or using computer, being tired in school, and having difficulties both in waking up and in sleeping (OR: 1.25; *p* = 0.011) [37].

We did not find any significant differences in the SVT among the children residing in rural versus urban areas. This is contrary to some studies that did report such differences [30,31,37]. This may be due to 100% penetration of internet services in Ujjian District’s rural areas and high phone possession in India in general [15].

In our study, the children who did not read books other than textbooks had an increased risk of excessive SVT (Table 3). Two studies in the USA reported that significant SVT distracts students from academic activities such as studying and doing homework, which can lead to learning and attention deficits and negative attitudes toward attending school [38,39]. Children who did not play outdoors for more than 2 h per day exhibited excessive SVT in the present study. Children with significant SVT spend less time in outdoor activities or playing and can have a sedentary lifestyle, which can lead to childhood obesity [40].

Headache is significantly associated with mobile phone usage. One study reported the associations between computer use and health problems in students and found that the prevalence of headache was 51% among girls and 24% among boys [41]. Children watching TV for more than 3 h reported having headache more frequently than those watching TV for less than 2 h [42]. Excessive use of mobile phones is considered a risk factor particularly for the development of migraine [42]. Another study reported an association of computer use with headache and neck pain among adolescent school students in a resource-poor country [43].

Digital device use has been associated with symptoms of dry eyes and tear film instability. Tear film instability increases with electronic device use during focused SVT, which leads to an increased interblink interval, leading to eye fatigue and tear film instability [4,12,14]. Blue light emitted from smart mobile device screens is also associated with eye fatigue and poor sleep quality [44]. Late-night screen viewing activity leads to increased sympathetic activity, which leads to decreased tear formation and secretion [12,44]. Increased sympathetic activity and decreased tear formation also affect sleep quality [12,44]. We speculate that eye pain had a negative effect on SVT in our study. Dry eye symptoms, such as eye fatigue or dry eye sensation, have a negative effect on daily life activities among adults [45].

The present study has some limitations. The definition of the outcome variable, excessive SVT, varies among studies and therefore our results cannot be strictly compared with other studies. The questionnaire used in the study was face-validated and content validity index was calculated; however, other validation steps were not carried out, which is a major limitation. The present study is cross-sectional in nature; therefore, interpretations regarding causality between the outcome variable, excessive SVT, and independent variables cannot be made. A bidirectional relationship between the independent and dependent variables cannot be ruled out. The study was conducted after the first wave of the COVID-19 pandemic; most children were receiving online education during this period, which might have provided greater opportunities for children to use online platforms for noneducational activities. Additionally, all data were self-reported, which might have introduced various biases, such as the social desirability bias, recall bias, and confirmation bias. We could not collect anthropometric measurement data on the children, which would have provided more robust evidence for the correlation of SVT with obesity. One must be careful in generalizing the findings of the present study due to the specific timing (post-COVID-19 pandemic), the location of data collection, and the wide age range of data collected.

## 5. Conclusions

The prevalence of excessive SVT in children aged 3 to 15 years was 18%. Thus, in our region, excessive SVT has the potential to affect the physical health of children. Mobile phone use before bedtime and early in the morning, the presence of a TV in bedroom, and disinterest in reading books other than textbooks were identified as potential risk factors for excessive SVT in the study. Further research is warranted to validate these associations.

## Figures and Tables

**Figure 1 ijerph-20-03449-f001:**
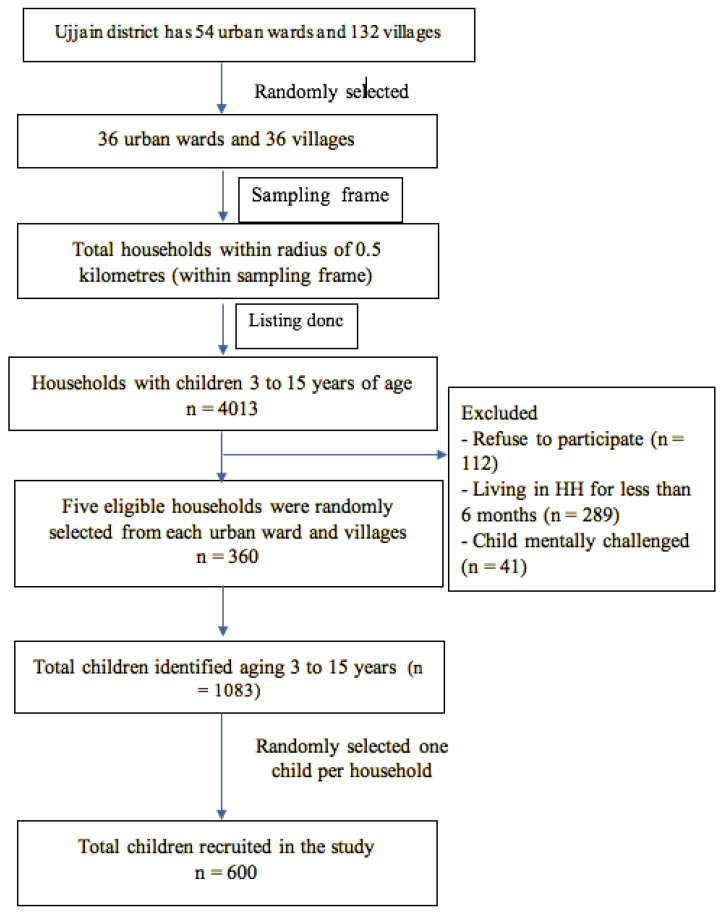
Sampling procedure followed in the study.

**Table 1 ijerph-20-03449-t001:** Bivariate analysis of sociodemographic factors associated with <2 h/day of screen viewing time in the 600 children included in the study.

Variable	Total n = 600	Screen Viewing Time (<2 h/Day)		OR	95% CI	*p* Value
		Yes (n = 108)	No (n = 492)			
**Gender**						
Girls	314 (52)	68 (22)	246 (78)	R	R	-
Boys	286 (48)	40 (14)	246 (86)	1.7	1.10–2.60	0.015
**Place of residence**						
Urban	307 (51)	59 (19)	248 (81)	R	R	-
Rural	293 (49)	49 (17)	244 (83)	1.18	0.77–1.79	0.427
**Age in years**						
≤6	245 (41)	82 (33)	163 (67)	R	R	-
>6	355 (59)	26 (7)	329 (93)	0.15	0.97–0.25	<0.001
**Type of family**						
Joint	196 (33)	28 (14)	168 (86)	Ref	Ref	-
Nuclear	404 (67)	80 (20)	324 (80)	0.67	0.42–1.07	0.1
**Number of family members**						
3–5	452 (75)	85 (19)	367 (81)	Ref	Ref	-
>5–9	116 (19)	19 (16)	97 (84)	1.18	0.68–2.03	0.547
>10	32 (5)	4 (13)	28 (87)	1.62	0.55–4.74	0.378
**Overcrowding**						
Yes	226 (38)	31 (14)	195 (86)	Ref	Ref	-
No	374 (62)	77 (21)	297 (79)	0.61	0.38–0.96	0.035
**Mother’s level of education**						
Uneducated/primary	251 (42)	53 (21)	198 (79)	Ref	Ref	-
Up to high school	141 (24)	21 (15)	120 (85)	1.52	0.87–2.66	0.133
Graduate/PG	208 (35)	34 (16)	174 (84)	1.36	0.85–2.20	0.193
**Mother’s occupation**						
Unemployed	296 (49)	61 (21)	235 (79)	Ref	Ref	-
Labourer	169 (28)	21 (12)	148 (88)	1.80	1.06–3.12	0.027
Salaried job	135 (23)	26 (16)	109 (81)	1.08	0.65–1.81	0.746
**Father’s level of education**						
Uneducated/primary	91 (15)	15 (16)	76 (84)	Ref	Ref	-
Up to high school	318 (53)	65 (20)	253 (80)	0.76	0.41–1.42	0.402
Graduate/PG	191 (32)	28 (15)	163 (85)	1.14	0.57–2.27	0.691
**Father’s occupation**						
Unemployed	27 (5)	6 (22)	21 (78)	Ref	Ref	-
Self employed	196 (33)	36 (18)	160 (82)	1.21	0.47–3.37	0.48
Labourer	203 (34)	33 (16)	170 (84)	1.47	0.55–3.92	0.77
Salaried job	174 (29)	33 (19)	141 (81)	1.22	0.45–3.26	0.40

**Table 2 ijerph-20-03449-t002:** Bivariate analysis of association of type of screen time, its duration, and child-related health factors with the outcome of <2 h/day of screen viewing in the 600 children included in the study.

Variable	Total n = 600	Screen Viewing Time (<2 h/Day)		OR	95% CI	*p* Value
		Yes (n = 108)	No (n = 492)			
**Night sleep time**						
≥6 h	537 (90)	104 (19)	433 (81)	R	R	-
<6 h	63 (10)	4 (6)	59 (94)	3.54	1.25–9.97	0.017
**Mobile phone use at bedtime**						
No	218 (36)	79 (36)	139 (64)	R	R	-
Yes	382 (64)	29 (8)	353 (92)	6.91	4.32–11.05	<0.001
**TV in bedroom**						
No	187 (31)	97 (52)	90 (48)	R	R	-
Yes	413 (69)	11 (3)	402 (97)	39.38	20.27–76.52	<0.001
**Morning mobile screen time**						
No	203 (34)	73 (36)	130 (64)	R	R	-
Yes	397 (66)	33 (8)	364 (92)	5.80	3.70–9.10	<0.001
**Child habituated to screen**						
No	293 (49)	94 (32)	199 (68)	R	R	-
Yes	307 (51)	12 (4)	295 (74)	9.88	5.48–17.82	<0.001
**Child reads books**						
Yes	332 (55)	86 (26)	246 (74)	R	R	-
No	268 (45)	20 (7)	248 (93)	3.90	2.36–6.44	<0.001
**Story telling by parents**						
Yes	237 (40)	69 (29)	168 (71)	R	R	-
No **Outdoor play**	363 (60)	37 (10)	326 (90)	3.41	2.20–5.26	<0.001
Yes	430 (72)	89 (21)	343 (79)	R	R	-
No	170 (28)	19 (11)	151 (89)	2.07	1.21–3.52	0.007
**Outdoor play ≥ 2 h**						
Yes	244 (41)	72 (30)	172 (70)	R	R	-
No	356 (59)	35 (10)	321 (90)	4.12	2.63–6.44	<0.001
**Headache**						
No	483 (81)	100 (21)	383 (79)	R	R	-
Yes	117 (19)	8 (7)	109 (93)	3.31	1.67–7.53	0.001
**Eye pain**						
No	478 (80)	96 (20)	382 (80)	R	R	-
Yes	122 (20)	12 (10)	110 (90)	2.30	1.21–4.53	0.010
**Eye itching**						
No	417 (70)	102 (29)	315 (76)	R	R	-
Yes	183 (30)	6 (3)	177 (97)	9.52	4.10–22.20	<0.001

**Table 3 ijerph-20-03449-t003:** Multivariate analysis of factors associated with >2 h/day of screen viewing time in 600 children included in the study with adjusted odds ratios (a OR).

Variable	a OR	95% CI	*p* Value
**Girls vs. boys**	1.31	0.58–2.97	0.512
**Age (continous variable)**	1.63	1.39–1.91	<0.001
**Mobile phone use at bedtime**	3.35	1.46–7.69	0.004
**Child habituated to screen according to parent’s perception**	8.46	2.77–25.96	<0.001
**TV in bedroom**	35.91	13.30–96.94	<0.001
**Morning mobile screen time**	6.40	2.76–14.84	<0.001
**Child not reading books other than textbooks**	6.45	2.35–17.73	<0.001
**Less than 2 h/d of outdoor play**	5.17	2.24–11.95	<0.001
**Eye pain present**	0.13	0.02–0.68	0.012

## Data Availability

The data presented in this study are available on request from the corresponding author. Please mention Shilpa_SVT_dataset when requesting access to the data. The data are not publicly available due to the sensitive nature of the data provided, as perceived by the parents.

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
