# Peer review of "Prevalence and Determinants of Excessive Screen Viewing Time in Children Aged 3–15 Years and Its Effects on Physical Activity, Sleep, Eye Symptoms and Headache"

_ijerph, 2023, doi:10.3390/ijerph20043449_

Round 1

Reviewer 1 Report (Previous Reviewer 1)

(OR: 163., P < 0.001);   typo ?

This is a cross-sectional study conducted in a community not a community-based study per se

“A questionnaire was developed through a review of literature and by two subject experts.” The validity of the questions needs be included and cited. 

“validated in a pilot study done among 30 children” results of this need to be included and cited. 

“Multivariate logistic regression was used to examine the association of potential risk factors with screen time 127 after adjusting for other covariates such as age and sex.”

This is vague, the precise covariates need to be listed in detail (continuous, categorical, cutpoints) as related to each IV in table 3. 

To many citations are used, generally 30~35 is the limit for a small cross-sectional study. 

Author Response

Comments and Suggestions for Authors

Question 1 (OR: 163., P < 0.001);   typo ?

Reply: yes, it was a typo. Corrected

Question 2 This is a cross-sectional study conducted in a community not a community-based study per se

“A questionnaire was developed through a review of literature and by two subject experts.” The validity of the questions needs be included and cited. 

“validated in a pilot study done among 30 children” results of this need to be included and cited. 

Reply: We have now added the following in the methods section:

“The questionnaire was face validated in a pilot study done among 30 children. Full age range of 3-15 years was tested in the pilot study and the questionnaire was found to be appropriate for use. The content validity index of the questionnaire was 0.87.

It is against tradition to include results of pilot study in the main study, so the authors have decided not to include the results of pilot in the main study.

Question 3 “Multivariate logistic regression was used to examine the association of potential risk factors with screen time 127 after adjusting for other covariates such as age and sex.”

This is vague, the precise covariates need to be listed in detail (continuous, categorical, cutpoints) as related to each IV in table 3. 

Reply: We have now added the following in the methods section:

Stepwise multivariate logistic regression with backward elimination was used to develop the final model. In bivariate analyses, a P value of < 0.1 was considered significant for entry into the model, with excessive SVT as outcome variable. The independent variables were: girls versus boys; place of residence (urban versus rural); age in years (less than and equal to 6 versus more than 6 years); type of family (joint versus nuclear); number of family members (3 to 5 versus >5 to 9 versus more than 10); overcrowding ()yes versus no); mother’s education (uneducated/ till primary school versus up to high-school versus graduate/post graduate); mother’s occupation (unemployed versus labourer versus salaried job); father’s education uneducated/ till primary school versus up to high-school versus graduate/post graduate); father’s occupation (unemployed versus self-employed versus labourer versus salaried job). The independent variables for child related health factors were: night sleep time (more than or equal to 6 hours versus less than 6 hours); mobile phone use at bed time (no versus yes); TV use at bed time (no versus yes); morning mobile screen time (no versus yes); child habituated to screen (no versus yes); child read books other than school books (yes versus no); story telling by parents (yes versus no); outdoor play (yes versus no); outdoor play more than or equal to 2 hours (yes versus no); headache (no versus yes); eye pain (no versus yes); eye itching (no versus yes).

The backward elimination was repeated till the P value of all the predictor variables was less than 0.1 except age and sex; we designated this as final model. Adjusted odds ratios and their respective 95% confidence intervals (CI) were then calculated from the final model. A P value of <0.05 was considered significant in the final model.

Question 4 To many citations are used, generally 30~35 is the limit for a small cross-sectional study. 

Reply: There is no limit to number of citations according to the journal. The number of citations increased according to the comments of previous reviewers, as clarity on certain points in the discussion was needed.

Round 2

Reviewer 1 Report (Previous Reviewer 1)

Just saying the SVT questionnaire is validated does not provide validation. The analysis needs to be shown in the supplement, or you need a citation showing evidence of validation. It’s still a limitation.

Table 3. Multivariate analysis of factors associated with screen viewing time <2 hours/day in 600 children included in the study with adjusted odds ratios (OR). Is this supposed to read >2hrs per day? Children with a TV in their bedroom are 35.91 times more likely to get < 2 hrs. per day of SVT? Is this correct? Please check throughout. 

Author Response

Reply to reviewer

 Question 1 Just saying the SVT questionnaire is validated does not provide validation. The analysis needs to be shown in the supplement, or you need a citation showing evidence of validation. It’s still a limitation.

Reply: We have written the following in methods section

The questionnaire was face validated in a pilot study done among 30 children. Full age range of 3-15 years was tested in the pilot study and the questionnaire was found to be appropriate for use. The content validity index of the questionnaire was 0.87. We have now provided the calculation of the CVI as follows in the appendix.

Content Validity Ratio Calculation for the SVT questionnaire

Question no

 Experts 1

Experts 2

Experts 3

CVR

(NE − N/2) / (N/2)

2.1

×

0.33

2.2

1

2.3

1

2.4

×

0.33

2.5

1

2.6

1

2.7

1

2.8

1

2.9

1

2.10

1

2.11

×

0.33

2.12

1

2.13

1

2.14

1

2.15

1

2.16

×

0.33

2.17

1

2.18

1

2.19

1

2.20

1

17.33

Content Validity Ratio = (NE − N/2) / (N/2)

  • NE = Number of SME panelists indicating “essential”
  • = Total number of SME panelists

CVR = (0.33+1+1+0.33+1+1+1+1+1+1+0.33+1+1+1+1+0.33+1+1+1+1)

Total CVR = 17.33

Content Validity Index = (CVR) / Total number of questions

CVI = (0.33+1+1+0.33+1+1+1+1+1+1+0.33+1+1+1+1+0.33+1+1+1+1) / 20

CVI = 0.867

Additionally we have added the following in the limitations of the study:

The questionnaire used in the study was face validated and content validity index was calculated, however other validation steps were not done, which is a major limitation.

Question 2 Table 3. Multivariate analysis of factors associated with screen viewing time <2 hours/day in 600 children included in the study with adjusted odds ratios (OR). Is this supposed to read >2hrs per day? Children with a TV in their bedroom are 35.91 times more likely to get < 2 hrs. per day of SVT? Is this correct? Please check throughout. 

Answer: Thank you for pointing out the typo, which we have corrected now.

This manuscript is a resubmission of an earlier submission. The following is a list of the peer review reports and author responses from that submission.

Round 1

Reviewer 1 Report

  1. Summary of the research

The authors investigate the association of prolonged SVT and possible determinates and health outcomes in a moderately sized locally representative pediatric population. The study aims to provide initial evidence of a potential problem that has been documented in the literature in other countries. Potential determinates were identified. Although cross-sectional in nature some of the ORs are very high, producing several potential hypotheses and the need for further research.

The primary limitation of the study is the cross-sectional design therefore causal inference cannot be made. The association of risk factors with the outcome variable may be bidirectional.

Major issues

The night-sleep <6 hours is not correct. It’s not clear why this cut point was used. Sleep time is age based. https://pubmed.ncbi.nlm.nih.gov/29073412/

https://www.sleepfoundation.org/children-and-sleep/how-much-sleep-do-kids-need

<10 hrs. is the cut point for this population. Please update this.

Because of the cross-sectional design, interpretations about causality between SVT and the determinates cannot be made. Therefore, a bidirectional relationship between the independent and dependent variables cannot be ruled out. Need to state this in the limitations.

The findings are limited to the local region and not India as a whole, this should be made clear in the limitations.

The survey does not appear to be validated, this is limitation and should be stated.

Minor issues

Potential mechanisms with the associations should be discussed in one paragraph.

This is not a cohort study; the term incidence should not be used.

The conclusion is written using casual language. Please modify it to language of association and that further research needs to be done validate these associations.

Author Response

Reply on comments and Suggestions by Reviewer 1.

  1. Summary of the research

  • The authors investigate the association of prolonged SVT and possible determinates and health outcomes in a moderately sized locally representative pediatric population. The study aims to provide initial evidence of a potential problem that has been documented in the literature in other countries. Potential determinates were identified. Although cross-sectional in nature some of the ORs are very high, producing several potential hypotheses and the need for further research.

Reply           The authors thank the reviewer for reviewing the article.

  • The primary limitation of the study is the cross-sectional design therefore causal inference cannot be made. The association of risk factors with the outcome variable may be bidirectional.

Reply           This is general drawback of cross-sectional studies and authors accept the point raised by the reviewer. However, we believe that this study will provide base of future studies using more robust study design.

                     The authors have tried to be careful in the interpretation of results due to cross-sectional design of the study.

We have added the following in the discussion section under limitations of the study:

                     “The present study is cross-sectional in nature, therefore interpretations regarding causality between the outcome variable excessive SVT and independent variable cannot be made. A bidirectional relationship between the independent and dependent variables cannot be ruled out. ”

Major issues

  • The night-sleep <6 hours is not correct. It’s not clear why this cut point was used. Sleep time is age based. https://pubmed.ncbi.nlm.nih.gov/29073412/

https://www.sleepfoundation.org/children-and-sleep/how-much-sleep-do-kids-need

Reply           The night sleep time was selected as less than 6 hours so as to identify chronic short sleep which is defined as mean nocturnal sleep duration of less than 6 hours. https://pubmed.ncbi.nlm.nih.gov/31575322/ However, the above study was done in adults and we have now repeated the analysis using National Sleep Foundation's sleep duration recommendations and used >10 hours sleep duration for the analysis. The new results are depicted in Table 2.

  • <10 hrs. is the cut point for this population. Please update this.

Reply           We have revised the results in Table 2 according to cut-off  of >10 hours.   

  • Because of the cross-sectional design, interpretations about causality between SVT and the determinates cannot be made. Therefore, a bidirectional relationship between the independent and dependent variables cannot be ruled out. Need to state this in the limitations.

Reply           We have added the following in the discussion section under limitations of the study:

                     “The present study is cross-sectional in nature, therefore interpretations regarding causality between the outcome variable excessive SVT and independent variable cannot be made. A bidirectional relationship between the independent and dependent variables cannot be ruled out. ”

  • The findings are limited to the local region and not India as a whole, this should be made clear in the limitations.

Reply           We have added the following in the discussion section under limitations of the study:

                     “The findings of the study can be generalized to similar settings in India and elsewhere.”

  • The survey does not appear to be validated, this is limitation and should be stated.

Reply           We did face validation of the questionnaire in a pilot study. We have now added the following in the methods section:

                     “The questionnaire was face validated in a pilot study done among 30 children. The result of the pilot study were not included in the final results. Minor changes to the questionnaire were made after the pilot study.”

Minor issues

  • Potential mechanisms with the associations should be discussed in one paragraph.

Reply           Done

  • This is not a cohort study; the term incidence should not be used.

Reply           Thank you for pointing this out. We have changed it to prevalence throughout the manuscript.

  • The conclusion is written using casual language. Please modify it to language of association and that further research needs to be done validate these associations.

Reply           The conclusions are now rewritten:

“The prevalence of excessive SVT in children aged 3 to 15 years was 17.8% in Ujjian district. Thus, in our region excessive SVT has the potential to affect physical health of children. Mobile phone use before bedtime and in early morning, presence of TV in bedroom, and disinterest in reading books other than textbooks, were identified as potential risk factors for excessive SVT in the study. Further research is warranted to validate these associations.”

Reviewer 2 Report

Comments

Why the age group was not divided - children aged 3-6 spend their time differently than teenagers

If the research was carried out in the city and in the countryside why not compare it

The period May - August 2021 is the time of a pandemic - did children (especially school-age children) not have remote classes - this information is missing at work, and should appear in the introduction

The authors in the discussion refer to the results of this work, which were not presented in the results section (line: 156-157; 176-177)

no new publications relating to the problem that the authors pay attention to at the very end of the work - the Covid 19 pandemic

these are just sample articles:

  1. Eyimaya, Aslihan Ozturk, and Aylin Yalçin Irmak. "Relationship between parenting practices and children's screen time during the COVID-19 Pandemic in Turkey." Journal of pediatric nursing56 (2021): 24-29.
  2. Bergmann, Christina, et al. "Young children’s screen time during the first COVID-19 lockdown in 12 countries." Scientific reports1 (2022): 1-15.
  3. Lau, Eva Yi Hung, and Kerry Lee. "Parents’ views on young children’s distance learning and screen time during COVID-19 class suspension in Hong Kong." Early Education and Development6 (2021): 863-880.
  4. McArthur, Brae Anne, et al. "Recreational screen time before and during COVID‐19 in school‐aged children." Acta Paediatrica (Oslo, Norway: 1992)(2021).
  5. Musa, Sarah, Rowaida Elyamani, and Ismail Dergaa. "COVID-19 and screen-based sedentary behaviour: Systematic review of digital screen time and metabolic syndrome in adolescents." PloS one3 (2022): e0265560.
  6. Chwałczyńska, Agnieszka, and Waldemar Andrzejewski. "Changes in body mass and composition of the body as well as physical activity and time spent in front of the monitor by students of the Wroclaw University of Health and Sport Sciences during the period of COVID-19 restrictions." International Journal of Environmental Research and Public Health15 (2021): 7801.

Author Response

Reply to Comments and Suggestions by Reviewer 2.

  • Why the age group was not divided - children aged 3-6 spend their time differently than teenagers?

Reply              We are not sure what the reviewer means here. The age was divided in 3 categories in Table 1 to show association of age with outcome variable excessive SVT. However, the age was taken as continuous variable in the multivariate logistic regression analysis as shown in Table 3. This is done as per tradition of taking age as a continuous variable in the final multivariate logistic regression model. The age is significantly associated with the outcome, which will not change even if we use age categories.

  • If the research was carried out in the city and in the countryside why not compare it?

Reply              The rural and urban children’s SVT was compared and presented in table 1. However, the association was not found to be statistically significantly correlated. Therefore, it was not discussed in the discussion section.

  • The period May - August 2021 is the time of a pandemic - did children (especially school-age children) not have remote classes - this information is missing at work, and should appear in the introduction

Reply.            The study was planned before the Covid pandemic. We have now discussed the impact of pandemic in the discussion section as follows:

It was not the objective of the present study to look at the effects of the Covid-19 pandemic on the SVT as out study was planned before the pandemic. However, the pandemic induced lockdown did have effect on SVT of both parents and children [23, 24]. Increase TV and multimedia mobile use was observed [25]. Schools started distance learning classes, thus increasing SVT for education [25]. Increased SVT leading to decreased physical activity was also observed among adolescents [26].

  • The authors in the discussion refer to the results of this work, which were not presented in the results section (line: 156-157; 176-177)

Reply.           The results mentioned by the reviewer are presented in Table 1, we have now added Table 1 at the end of the sentence for ease of reading.

  • No new publications relating to the problem that the authors pay attention to at the very end of the work - the Covid 19 pandemic

these are just sample articles:

Reply        We have now discussed the impact of pandemic in the discussion section as follows:

It was not the objective of the present study to look at the effects of the Covid-19 pandemic on the SVT as out study was planned before the pandemic. However, the pandemic induced lockdown did have effect on SVT of both parents and children [23, 24]. Increase TV and multimedia mobile use was observed [25]. Schools started distance learning classes, thus increasing SVT for education [25]. Increased SVT leading to decreased physical activity was also observed among adolescents [26].

Reviewer 3 Report

Thank you for giving me the opportunity to review your paper. This is an interesting topic and one of growing import in preventing the problems caused by use of modern technology.

There are some major revisions that need to be considered:

The definition of the independent variable is problematic. The variable of time spent in front of screens (ST) is dichotomised, with excessive ST as greater than 2 hours. There is no discussion about the reason for this choice. There needs to be a robust discussion about the validity of choosing 2 hours, as the statistics depend on the definition of this variable.

In the literature, the definition of excessive ST varies considerably. For instance, Çaylan, Nilgün et al. “Associations Between Parenting Styles and Excessive Screen Usage in Preschool Children.” Turk Pediatri Arsivi 56.3 (2021): 261–266 defined excessive screen exposure as being more than 4 hours. Philip Baiden, et al., (The association between excessive screen-time behaviours and insufficient sleep among adolescents: Findings from the 2017 youth risk behaviour surveillance system, Psychiatry Research, Volume 281,2019), dichotomise the time spent on screens to less than 3 hours and 3 or more hours. The American Association of Paediatrics guidelines state that children aged < 2 should not have any ST, while those aged 2–5 years should be limited to 1 hour.

Consequently, when comparing the findings in this paper to other papers it must be noted that definitions of excessive ST are different in various papers. This limits the ability to make comparisons and draw conclusions about reproducibility. More discussion about this needs to be included in the limitations of the study, as well as pointing out the need to standardise the definition of excessive ST.

There are some minor revisions also noted:

  • Line 32, “Recent data suggests that children… data is plural therefore it should be suggest.
  • Line 39-40, “the recommended limit set by the WHO on PA per day has not been achieved” … requires a reference
  • Line 42, “spend at least 4 h in screen viewing daily.” …needs a reference.
  • Line 42, “Almost all school-going children watch TV, and one thirds of young children play games on computers or electronic devices.” …Is this in the USA? If so, the sentence needs to be connected to the previous sentence, and again a reference is needed to support the statement.
  • Line 70, “In a study in Karamsad, Gujrat, India, 17% children had SVT”. This is not what the study reports. The study concludes that 17% of children had less than 1 hour SVT.
  • Line 127, “An equal number of urban and rural children was included.” It is not exactly equal (307,293). A more accurate statement would be approximately equal.
  • Line 162, “reported the SVT for girls and boys to be 2.8”, does this mean average, median maximum? Please clarify.
  • Line 164, Variables like age need to be clarified, e.g., increasing age is correlated with …
  • Line 218, “We believe that the children who read books other than textbooks have lesser SVT than those who do not read such books.” As a principle, in scientific papers, it is best not to express beliefs, but implications and conclusions based on literature or evidence, such as is quoted in the next sentence.

Author Response

Reply to Comments and Suggestions for Authors by Reviewer 3

  • Thank you for giving me the opportunity to review your paper. This is an interesting topic and one of growing import in preventing the problems caused by use of modern technology.

Reply   Thank you for finding our paper interesting and for reviewing our paper.

There are some major revisions that need to be considered:

  • The definition of the dependent variable is problematic. The variable of time spent in front of screens (ST) is dichotomised, with excessive ST as greater than 2 hours. There is no discussion about the reason for this choice. There needs to be a robust discussion about the validity of choosing 2 hours, as the statistics depend on the definition of this variable.

Reply              We have now added the reason for choosing excessive SVT as two hours in the methods section. The cut-off of two hours is as per the Indian Academy of Pediatrics (IAP) guidelines for Indian children. We have added the following in the methods section:

Excessive SVT was defined as the screen activity of >2 h/day on any device, including TV screen, computers (desktop computer and laptops), tablets like iPads or Samsung Galaxy tabs, and mobile phones as per Indian Academy of Pediatrics guidelines [20]. The definition of the excessive SVT was defined post-hoc as the guidelines were not available at the time of start of the study.  

  • In the literature, the definition of excessive ST varies considerably. For instance, Çaylan, Nilgün et al. “Associations Between Parenting Styles and Excessive Screen Usage in Preschool Children.” Turk Pediatri Arsivi3 (2021): 261–266 defined excessive screen exposure as being more than 4 hours. Philip Baiden, et al., (The association between excessive screen-time behaviours and insufficient sleep among adolescents: Findings from the 2017 youth risk behaviour surveillance system, Psychiatry Research, Volume 281,2019), dichotomise the time spent on screens to less than 3 hours and 3 or more hours. The American Association of Paediatrics guidelines state that children aged < 2 should not have any ST, while those aged 2–5 years should be limited to 1 hour.

Reply.             We agree with the reviewer that in the literature, the definition of excessive ST varies considerably. We have discussed this point in the discussion section now.

The definition of the outcome variable excessive SVT varies considerably in literature. One study [21] defined it as more than 4 hours, another one defined it as more than 3 hours [22] and Indian academy of Pediatrics defined it as 2 hours [20]. The American Association of Pediatrics guidelines state that children aged < 2 should not have any ST, while those aged 2–5 years should be limited to 1 hour [2]. Thus, the definition of excessive SVT should be kept in mind while making comparisons between studies and their generalizability and reproducibility. We, therefore, need a standardized definition of excessive SVT.

  • Consequently, when comparing the findings in this paper to other papers it must be noted that definitions of excessive ST are different in various papers. This limits the ability to make comparisons and draw conclusions about reproducibility. More discussion about this needs to be included in the limitations of the study, as well as pointing out the need to standardise the definition of excessive ST.

Reply              We have added the above point in the limitations:

“The definition of the outcome variable excessive SVT varies among studies and therefore, our results cannot be strictly compared with other studies.”

There are some minor revisions also noted:

  • Line 32, “Recent data suggests that children… data is plural therefore it should be suggest.

Reply.           Done

  • Line 39-40, “the recommended limit set by the WHO on PA per day has not been achieved” … requires a reference

Reply.           Done

  • Line 42, “spend at least 4 h in screen viewing daily.” …needs a reference.

Reply.           Done

  • Line 42, “Almost all school-going children watch TV, and one thirdsof young children play games on computers or electronic devices.” …Is this in the USA? If so, the sentence needs to be connected to the previous sentence, and again a reference is needed to support the statement.

Reply.           Done

  • Line 70, “In a study in Karamsad, Gujrat, India, 17% children had SVT”. This is not what the study reports. The study concludes that 17% of children had less than 1 hour SVT.

Reply.           We have added “less than 1 hour” at the end of the sentence.

  • Line 127, “An equal number of urban and rural children was included.” It is not exactly equal (307,293). A more accurate statement would be approximately equal.

Reply.           Done

  • Line 162, “reported the SVT for girls and boys to be 2.8”, does this mean average, median maximum? Please clarify.

Reply.           It is mean number of hours of ST. This has been added in the text now.

  • Line 164, Variables like age need to be clarified, e.g., increasing age is correlated with …

Reply.            We have added “increasing age” to the start of the sentence.

  • Line 218, “We believe that the children who read books other than textbooks have lesser SVT than those who do not read such books.” As a principle, in scientific papers, it is best not to express beliefs, but implications and conclusions based on literature or evidence, such as is quoted in the next sentence.

Reply.            We have removed the sentence and added:

 “In our study children who did not read books other than text books had an increased risk of SVT (Table 3).”

Reviewer 4 Report

The study addresses a relevant issue, the effects of screen time in children. Overall, the introduction reads well but the outcome variables can be justified more. Why were they chosen? Some information on the outcome variables can be found in the discussion. Please find some specific recommendations to enhance the methods and results section in the attached pdf. 

The discussion can be improved by further explaining and interpreting the results. Additionally, some studies used to compare results to need clarification/more information. Please find specific comments/suggestions for improvement in the attached PDF.

Author Response

Reply to comments and suggestions by Reviewer 4.

Page 1

  • What was the age of the children, this is very broad

Reply           We have added age in the title  

  • Please define the age of the children

Reply           We have added age in the title  and not in abstract due to word limit

  • Reference WHO guidelines might be more appropriate

Reply           Thank you for your suggestion. However, we wish to go with American Academy of Paediatrics guidelines for the given sentence

  • Do you have numbers and/or references?

Reply.          Added reference

.                   

  • Do you have references for this?

Reply           Added reference

Page 2:

  • Perhaps divide this into two sentences to make reading easier.

Reply           Sentence is now reconstructed as follows:

All most 90% of the studies included in a systematic review indicated that increased SVT is associated with adverse sleep outcomes [14].

  • Why did you choose these outcome measure specifically?

Reply           The outcome was selected on bases of literature review done at the start of the study.

  • I suggest adding this to the methods section on data collection of SVT.

Reply.        We have shifted the definition of the outcome variable to end of section 2.2 of the methods as per reviewers suggestions.

  • Was there a rationale for choosing these groups?

Reply            The reason for stratification is clarified in sampling procedure where we used three staged cluster sampling. Therefore,  3 age groups were chosen. Also, NFHS data (based on which sampling was done) is available till 15 years of age.

  • This is repetition?

Reply           We have modified the sentence as follows:

In this community-based study we selected participants at the household level by using the three-stage cluster sampling method.

  • How was this information acquired?

Reply            The information was acquired by house-to-house visits. This information is added in the methods section as follows:

Then, a census of households was conducted to identify households with children aged 3–15 years using house-to-house visit.

Page 3:

  • When did a parent complete the questionnaire and when the child?

Reply           Children between 10 to 15 years completed their questionnaire themselves. This is added in the methods section now.

  • Can you give some examples?

Reply           Questionnaire is attached with the manuscript.

  • Was this a validated questionnaire or did you create the questions yourself?

Reply            The questionnaire was face validated in a piolet study. We have added the following details in the manuscript methods section, page 3:

                     “The questionnaire was face validated in a pilot study done among 30 children. The results of the pilot study were not included in the final results. Minor changes to the questionnaire were made after the pilot study.”

  • How many household visits were there?

Reply           n = 1826. This is added to the manuscript.

Page 4:

  • This is very large age range? What was the mean and SD??

Reply           We have added the following in the text now

A total of 600 children including 314 (52%) girls and 286 (48%) boys were included, with the mean (± standard deviation) age 8.82 (±3.3) years (range of 3–15 years)

  • Overall: tables 1 and 2 are barely addressed in the text. I suggest to add this, especially for table 2.?

Reply           We have added in discussion as well as in results. .

Page 5: No comments

Page 6:

  • Perhaps rephrase the title (table 2) to make clear what the outcome variable was?

Reply           Done

Page 7:

  • Did this study look at similar ages? Was it cross-sectional as well?

Reply           In the beginning of discussion we have now discussed the definition of outcome and also the limitation is added that previously published studies can not be strictly compared with our study due to variability in screen view time cut-off:

The definition of the outcome variable excessive SVT varies considerably in literature. One study [21] defined it as more than 4 hours, another one defined it as more than 3 hours [22] and Indian academy of Pediatrics defined it as 2 hours [20]. The American Association of Pediatrics guidelines state that children aged < 2 should not have any ST, while those aged 2–5 years should be limited to 1 hour [2]. Thus, the definition of excessive SVT should be kept in mind while making comparisons between studies and their generalizability and reproducibility. We, therefore, need a standardized definition of excessive SVT.

We have also added the following to the limitations section:

The present study has some limitations. The definition of the outcome variable excessive SVT varies among studies and therefore, our results cannot be strictly compared with other studies.

Page 8:

  • Can you compare these results to your results? Were it similar studies? One study reports percentages, the other hours, how does this compare to your results?

Reply           Please see reply to comment 19

  • Can you explain this?

Reply           We have now explained the crowding out effect as follows:

The decreased SVT in Chinese senior high school children might be because of the pressure of studying and preparing for college entrance examinations, which may compel students to pay more attention and devote more time on study-related behavior, resulting in a “crowding-out effect” in the motivation crowding theory [28]. Crowding out means that when an individual is offered extrinsic incentives for certain kinds of behavior—such as promising rewards for accomplishing some task—can sometimes undermine intrinsic motivation for performing that behavior [30].

  • How is this study relevant if it's not the right age group?

Reply           We have deleted the study from discussion

  • Please elaborate on this. Perhaps this should be moved to the introduction?

Reply           We have not used socio-ecological model in our study.

  • Please add the ages of the children included in these studies.?

Reply           Done

  • Sounds like an interesting study but not quite sure about the relevance here. Perhaps use it for the introduction.

Reply           We would like to use this study in discussion as the introduction is quite lengthy.

  • Can you add some information about the study population?

Reply          Done

Page 9:

  • Can you explain this?

Reply      We have explained the bias as follows:

Additionally, all data were self-reported, which might have introduced various biases-like the social desirability bias, recall bias and confirmation bias

Round 2

Reviewer 1 Report

Table 2 is not updated to the recommended 10hrs. 

Reviewer 2 Report

The work is very interesting and takes up a very important topic. The authors took into account my earlier comments. References should be standardized